# Exploring immersion through a fMRI-compatible multi-finger handheld haptic display

**Joonsub Byun**[1], **Joonseon Hwang**[1], **Yong-An Chung**[2], **Hyeonseok Jeong**[3], **Jooyeon Kim**[4], **Keehoon Kim**[1]*

1 Department of Mechanical Engineering, POSTECH, Pohang, South Korea, 2 Department of Nuclear Medicine, Incheon St. Mary's Hospital, College of Medicine, The Catholic University of Korea, Seoul, Republic of Korea, 3 Ewha Brain Institute, Ewha Womans University, Seoul, South Korea, 4 Center for Bio-Imaging and Translational Research, Korea Basic Science Institute, Cheongju, South Korea

* khk@postech.ac.kr

## Abstract

This research presents the development and evaluation of a functional magnetic resonance imaging (fMRI)-compatible, handheld, multi-finger haptic display, designed to augment and quantitatively measure immersion within virtual reality (VR) environments via tactile feedback mechanisms. Incorporating advanced pneumatic actuators, the device is capable of delivering differentiated pressure feedback to individual fingers, thereby facilitating a more realistic interaction within VR scenarios. Comprehensive fMRI-compatibility assessments have been successfully conducted, verifying the device's operational integrity within a 3T functional magnetic resonance imaging (fMRI) setting. This ensures its applicability for integrated neuroscientific explorations. Alongside, a VR platform was implemented to integrate audiovisual stimuli with haptic feedback, aiming to create a fully immersive experience for participants during fMRI studies. Experiments involving human subjects have demonstrated marked cortical activations in brain regions associated with immersive experiences, attributed to interactions with the VR content via the proposed haptic display. These findings highlight the potential of the developed haptic system in both enhancing the immersive quality of VR environments and serving as a novel instrument for the quantitative analysis of immersion through neuroimaging. The study advances the field by offering a new avenue for the exploration of neural mechanisms underlying immersive VR experiences, facilitated by fMRI-compatible haptic feedback.

## Introduction

Virtual reality (VR) offers users an array of experiences by facilitating interactions with virtual environments, showcasing its versatility not only in digital storytelling and entertainment but also across diverse fields such as healthcare, education, and engineering. For example, virtual reality can be used to train surgeons or firefighters using haptic devices and has cost and risk reduction advantages compared to

**Data availability statement:** The data underlying the results presented in the study are available from Figshare (Component Compatibility: https://figshare.com/s/d330fb-cb7ac1616df670. Haptic Device Compatibility: https://figshare.com/s/498cfd822704af2227a4. Experiment1: https://figshare.com/s/1a393e-87a37849307c75. Experiment2: https://figshare.com/s/bd888c28aa174a40e4df).

**Funding:** This work was supported in part by Institute of Information & Communications Technology Planning & Evaluation (IITP, https://www.iitp.kr) grant funded by the Korea government(MSIT) (No.2021-0-00986, Development of Interaction Technology to Maximize Realization of Virtual Reality Contents using Multimodal Sensory Interface), in part by the National Research Foundation of Korea (NRF, https://www.nrf.re.kr) grant funded by the Korea government (MSIT) (No.2022M3C1A3081359), in part by the Pilot Project for Commercialization of Police Technology Public Research through the Commercializations Promotion Agency for R&D Outcomes funded by the National Police Agency and the Ministry of Science and ICT (No. 1711174175). The funders had no role in study design, data collection and analysis, decision to publish, or preparation of the manuscript.

**Competing interests:** The authors have declared that no competing interests exist.

reality [1,2]. Immersion acts as the cornerstone of these applications, granting users a sense of 'being' within the task environment. Despite technological advances, maintaining immersion in the virtual environment, especially during physical interaction, remains a challenge due to users' sensitivity to the distinctions between the real and virtual worlds.

Several key factors contribute to VR immersion, including visual fidelity, spatial audio, interactivity, and presence. Among these, haptic feedback is particularly crucial because it completes the sensory experience by providing tactile sensations that correspond to virtual interactions. Haptic display is an interface aiming to enable bilateral signal communications between human and computer, and thus to greatly enhance the immersion and interaction of VR systems [3]. Since current consumer VR experiences are predominantly generated through visual and auditory feedback [4], incorporating haptic elements addresses a critical gap by engaging the sense of touch, creating more complete and compelling virtual experiences.

While researchers are dedicated to enhancing immersion through haptic technologies, accurately evaluating it remains a challenge. Previous studies have used qualitative methods, such as surveys, to evaluate VR immersion [5,6]. However, qualitative methods are limited by their inability to measure user experience in real time or accurately express the user's state. Therefore, quantitative research on immersion is necessary but currently underdeveloped.

Functional magnetic resonance imaging (fMRI) presents a quantitative method to explore immersion by examining brain activity with high spatial resolution, uncovering the neurological underpinnings of immersion and aiding in the improvement of VR content quality. However, most studies have primarily focused on the visual and auditory aspects of immersion [7], neglecting the critical role of haptic feedback. Haptic feedback is a critical component of system immersion, as it heightens sensory realism—a key factor in eliciting the user's subjective sense of presence [8].

The emergence of immersive technologies and the metaverse has highlighted the importance of haptic feedback in the VR market, leading to innovative applications from full-body haptic suits to hand controllers. Despite its significance, quantitative studies on immersion induced by haptic feedback are scarce. Earlier research on the impact of realistic sensory feedback in VR showed brain activation patterns akin to those from real-world physical interactions, though such studies often provided limited kinesthetic and force feedback [9]. Quantifying haptic immersion as illustrated in Fig 1, possibly through fMRI-compatible haptic displays, could lead to significant breakthroughs in VR technology, enabling the comparison and evaluation of VR systems with haptic interfaces and assisting researchers in identifying methods to enhance the VR experience.

Furthermore, advancements in fMRI-compatible robotics linked with VR have opened new possibilities for interaction within VR environments. Notably, a 7-DOF (degrees of freedom) interface has been developed to enhance feedback on upper limb movements [10], marking a significant step towards more nuanced interaction capabilities in VR, albeit primarily focusing on upper limb dynamics. Beyond this, fMRI-compatible robots have been employed for various applications, including

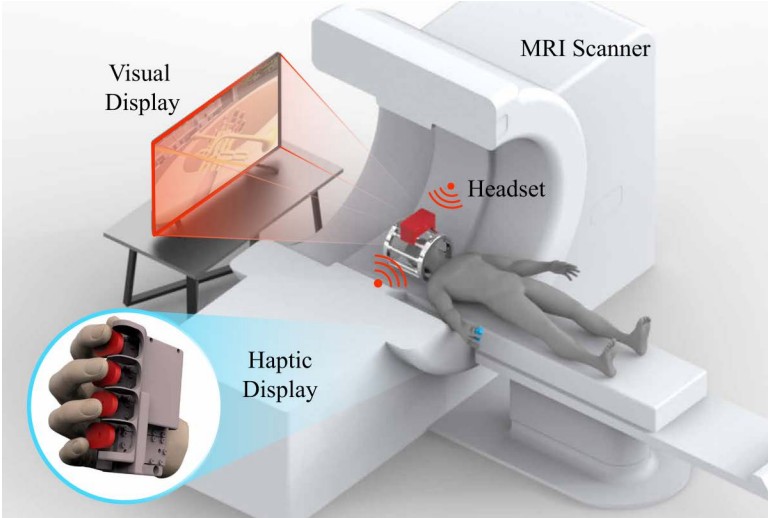

**Fig 1. fMRI-compatible virtual reality system with haptic interaction.**

rehabilitation and the study of motor control. For instance, a cable-driven haptic display has been proposed to facilitate research into precision pinch grip control [11], while devices like a 1-DOF hand rehabilitation device [12] and a 2-DOF finger exoskeleton [13] have been developed for hand function rehabilitation, specifically targeting hand flexion/extension tasks. Additionally, efforts to explore index finger flexion have led to the creation of a pneumatic stimulator [14], and a 3-DOF device has been proposed for more accurate hand movement tracking [15]. The diversity and potential applications of these fMRI-compatible haptic devices are systematically categorized in Table 1, providing a comprehensive overview of current capabilities and future directions in haptic feedback technology within VR systems.

Most fMRI-compatible robots utilized in fMRI studies are designed with simple, task-specific designs and limited degrees of freedom, restricting their use to specific, predefined tasks [16]. This necessitates the development of different devices for various tasks, underscoring the need for versatile, general-purpose haptic devices like HaptX [17], Senseglove [18] to evaluate users' immersion in a variety of virtual environments. In the context of fMRI studies, subject movement, particularly head movement, can degrade image quality [19]. Therefore, large arm movements that could affect the head's alignment and the RF coil are minimized, leading to the adoption of a multi-finger form factor for developing a haptic device that does not require upper limb movement. This multi-finger haptic device, characterized by its high degree of freedom, is universally applicable and capable of producing natural actions, offering a more natural and immersive user experience [20]. Glove-type haptic devices have the advantage of flexibility over handheld devices [21,22], but handheld haptic devices have the advantage of being held directly by the user, which makes them more stable during interaction. In contrast, most wearable devices require support by being attached to the hand using gloves or straps. In addition, handheld haptic devices can distribute the response force over a large area such as the palm, which can generate guidance force at the fingertips [23].

Building on this foundation, our research objectives are as follows: (1) Developing a multi-finger haptic display that ensures immersive VR experiences and fMRI-compatibility. (2) Exploring the feasibility of quantitative measurement regarding user immersion based on VR content suitable for multi-finger haptic displays. We adhere to the definition that immersion in VR is an objective level of sensory fidelity provided by a VR system [24]. In this context, immersion in VR primarily depends on sensory immersion, defined as "the degree to which the virtual simulation engages a range of sensory channels" [25]. Therefore, our research aims to provide a basis for quantifying

**Table 1. fMRI-compatible haptic devices with properties.**

| Ref. | Actuator | Handheld | Stimulation | DoF | with VR | |
|---|---|---|---|---|---|---|
| | | | | | with vision | with audio |
| [9] | P | No | Index finger | 3 | Yes | No |
| [10] | U | No | Upper limb | 7 | Yes | No |
| [11] | E | No | Thumb and index finger | 1 | No | No |
| [12] | P | No | Hand | 1 | Yes | No |
| [13] | U | No | Hand | 1 | No | No |
| [14] | P | Yes | Index finger | 1 | No | No |
| [15] | E | No | Upper limb | 3 | Yes | No |
| The proposed | P | Yes | Individual fingers | 4 | Yes | Yes |

P, Piezoelectric; U, Ultrasonic motor; E, Shielded electromagnetic motor.

immersion by measuring changes in activation within the brain's key sensory and cognitive networks. Specifically, we hypothesize that immersion levels will correlate with modulations in regions related to: (a) sensorimotor functions, such as the motor cortex, somatosensory cortices, and cerebellum; (b) audiovisual processing, like the temporal lobes; and (c) higher-order cognitive functions, including attention and sensory integration, governed by the frontal lobe.

## Materials and methods

### Design of multi-finger haptic display

Before developing our haptic device, we prioritized ensuring its safety and fMRI-compatibility. The use of ferromagnetic materials is avoided due to the strong magnetic field in MRI environments, which can attract such materials and cause accidents. Instead, we focus on using non-metallic materials like polymers and acrylics, or paramagnetic metals like aluminum and brass, which have low magnetic susceptibility. This approach aids in avoiding any safety hazards and ensures the device does not interfere with MRI image quality by generating electromagnetic interference or artifacts. We determined the suitability of electrical and mechanical components for MR environments through compatibility testing, which will be detailed in Section III.

Given the constraints of MR environments, we chose pneumatic actuators for our haptic device due to their high fMRI-compatibility. Piezoelectric actuators, while commonly used, often contain metal parts that pose compatibility challenges with MRI. On the other hand, pneumatic systems allow for separation from MRI-sensitive areas, offering higher compatibility and the potential for compact design. For the form factor, we opted for a handheld device over a glove to simplify use within the confined space of an MRI machine, eliminating the complexity of donning and doffing. We developed a pneumatic pressure display module that delivers force feedback to the user's four fingers, from the index to the little finger, by extending this technology. This multi-finger haptic display, when integrated with fMRI, enables the investigation of brain activation in response to haptic feedback for various VR tasks.

**Pressure display.** The interaction between users and virtual environments hinges on effective force feedback, necessitating a haptic device that provides individual pressure rendering for each finger. Current approaches, such as cable-driven systems for two fingers and ultrasonic motors for a single finger, are insufficient for fMRI-compatible environments where individual pressure feedback across multiple fingers is essential. Addressing this, we propose a novel pneumatic pressure display module.

## Multi-finger haptic display system

**Fabrication process of pressure display.** This module incorporates an acrylic pneumatic cylinder and a fiber optic sensor, designed to deliver accurate pressure to each finger. The acrylic parts are crafted using a carbon dioxide laser cutting machine, while silicone sheets within the cylinder are shaped with a diamond blade cutter, ensuring airtight seals. The assembly is modular, allowing customization to fit different hand sizes by adjusting the cylinder's length or thickness. The specific components and their assembly are detailed in Fig 2a, with complete specifications listed in Table 2.

**Construction details.** The module is constructed using a multi-layer assembly method. First, silicon gaskets are attached to both sides of a 100 $\mu$m-thick PET film. This film is then placed between acrylic plates with thicknesses of 2 mm, 3 mm, and 4 mm. This layered design serves two purposes: to reduce friction during movement and to ensure airtight sealing. Brass bolts and nuts fasten the structure, incorporating a 4 mm diameter pneumatic hose at the cylinder ends. The fiber optic sensor (Keyence FU-35FA), with its metal part removed for MRI compatibility, measures the end-effector's position. These customizable features allow the device to accommodate various hand sizes. The detailed fabrication process is illustrated in Fig 2b, showing the step-by-step assembly from initial layering to sensor installation.

By stacking the pressure display modules in parallel (Fig 3a) and attaching them to the back plate and separating the end-effectors of each module with a case (Fig 3b), we developed the handheld fMRI-compatible haptic display (Fig 3c). The proposed fMRI-compatible haptic display can deliver detailed pressure feedback from the index to the little finger. This setup is strategically designed to minimize MRI magnetic field interference, situating non-fMRI-compatible components, including the pneumatic pump, more than 7m away from the MRI bore. It features a pressure regulator (ITV2050-312N5, SMC) and dual solenoids per module for precise air control. An optical fiber amplifier (FS-V31M, Keyence) enhances sensor signals, integrating VR interactions seamlessly. System control and data acquisition rely on an NI-DAQ (USB 6343, National Instruments, TX, USA), ensuring comprehensive operation. The full system, optimized for use with a 3T MRI, is presented in Fig 4, underscoring our commitment to advancing immersive VR experiences in fMRI research.

## Performance of multi-finger haptic display

The selection of an open-loop control system was due to several design constraints inherent to a handheld, fMRI-compatible device. Specifically, the device requires compact sensors, but options that are simultaneously small and fMRI-compatible are rare. Although alternatives like fiber optic sensors exist, the fiber optic sensor (FS-V31M) is primarily intended for binary presence detection rather than continuous measurement. Consequently, their analog output exhibits significant non-linearity and variability depending on the specific device unit and surface conditions. These characteristics make it difficult to establish a consistent linear model required for precise closed-loop feedback. Alternatively, pressure can be estimated through a system model and used for control. However, conventional pressure estimation and control models do not function properly due to pressure attenuation and time delay caused by long transmission lines. This section identifies the performance characteristics of our haptic device systems, such as pressure drop along the transmission line, frequency response, and force transfer ratio. Furthermore, open-loop force control considering system characteristics is described.

For system analysis, the position and force at the end-effector of the pressure display module were measured. Position measurements were performed on a custom testbed made with a 3D printer. A linear strip (EM1-0-500-N, US-digital, WA, USA) was coupled to the end-effector of the pressure display module and measured using an optical encoder (EM1, US-digital, WA, USA) as depicted in Fig 5a. Force measurements were conducted using a custom testbed equipped with a force sensor (lcm 100, Futek, CA, USA) as shown in Fig 5b.

**System characterization.** The proposed pressure display module has a cylinder rod and upper and lower acrylic plates attached. Therefore, there is a trade-off relationship between the friction force of the cylinder rod and the sealing of the chamber. If the sealing is too tight, the rod does not move due to friction, and if the friction is reduced, the sealing becomes loose and pressure is not properly formed in the chamber. The pressure module is designed to allow for slight air leakage to adjust the trade-off.

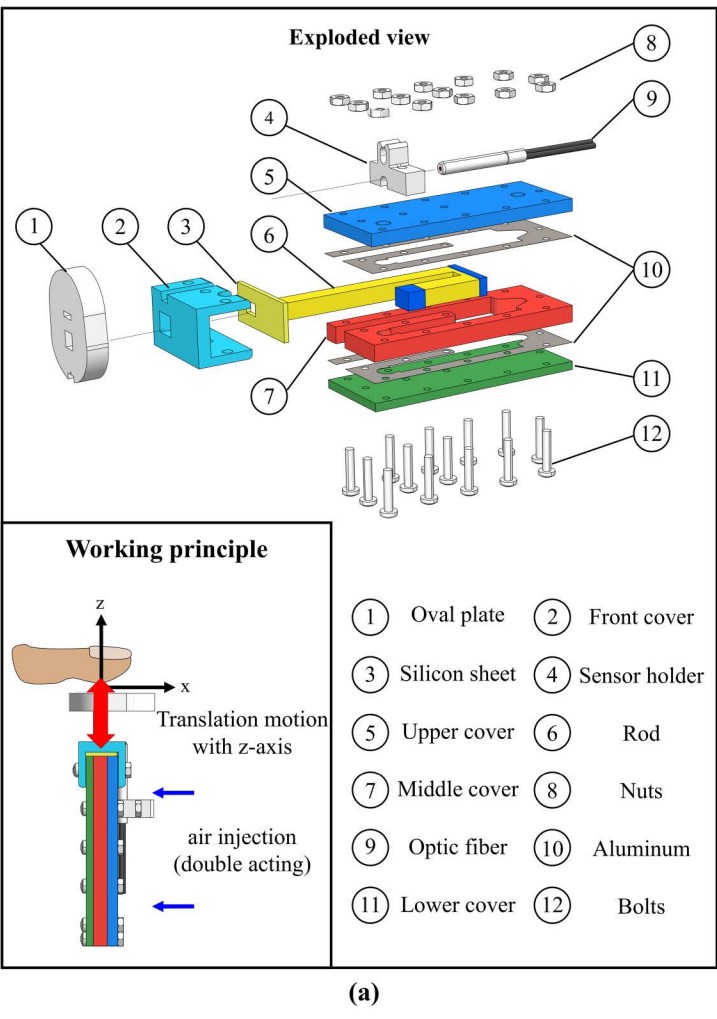

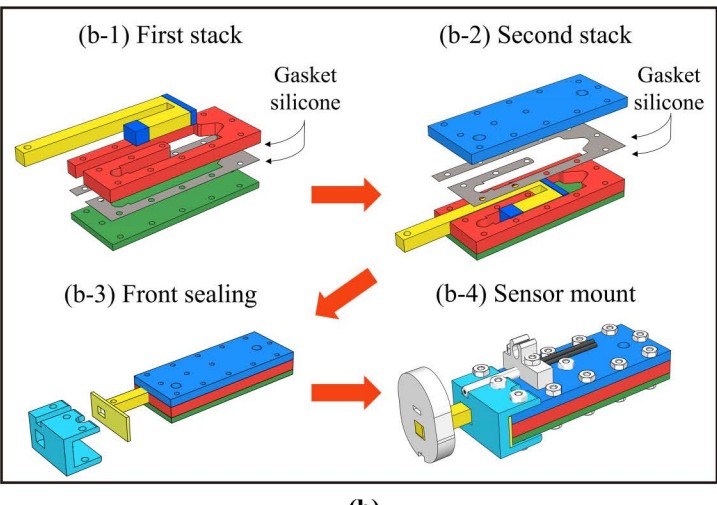

**Fig 2.** (a) Pressure display module exploded view and working principle. (b) Fabrication process of the pressure display module.

**Table 2. Pressure display module specification.**

| Specification | Value |
|---|---|
| Weight | 14.5 g |
| Size | $60 \times 21 \times 9$ mm$^3$ |
| Output force | <10 N |
| Static friction | < 1.1 N |
| Stroke | 13 mm |
| Source | Pneumatic |

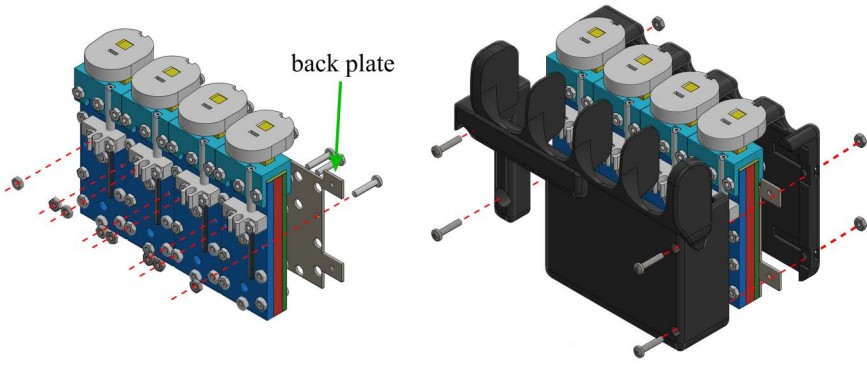

(a) Stack modules parallelly (b) Combine back plate and case

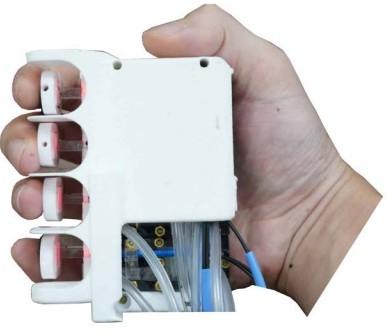

(c) Haptic display in hand

**Fig 3. fMRI-compatible multi-finger haptic display.**

An experimental system identification based on frequency analysis was applied to the pressure display module in order to describe the frequency range that can be operated. The frequency response was measured by commanding a sinusoidal pressure input to the front and back chambers of the pressure display module. The phase difference between the sinusoidal input to the front chamber and the sinusoidal input to the rear chamber was set at 90 degrees and measured for 20 frequencies between 0.1 and 3 Hz. Because the air leakage amounts of the front and rear chambers are different, the magnitude ratio of the two input signals was set to 1.8, which is the ratio found experimentally. The bode plot, which is the result of system identification, is shown in Fig 6. Investigation of frequency response analysis showed that the bandwidth was found to be 1.5 Hz, and phase delay occurs due to a long transmission line.

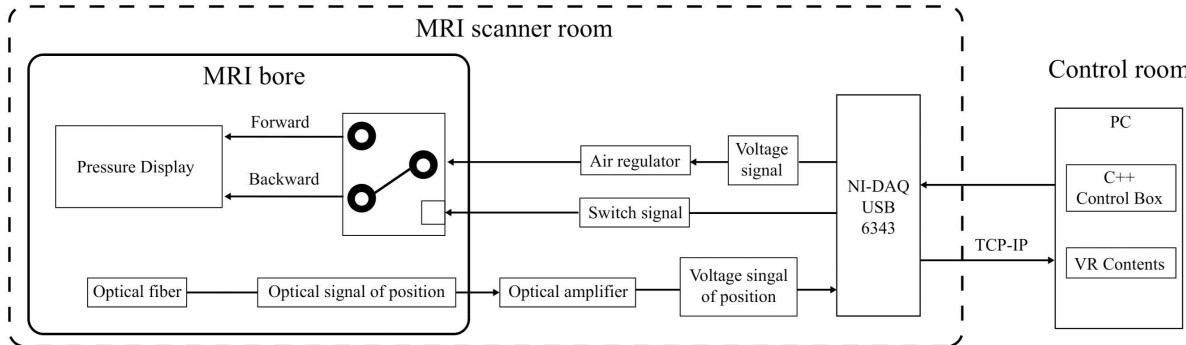

**Fig 4. System diagram of the proposed multi-finger haptic display.**

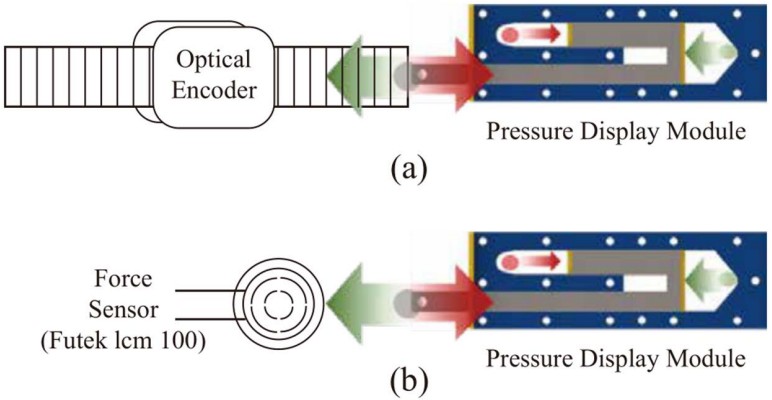

**Fig 5.** (a) Position measurement test bed with optical encoder and pressure display module. (b) Force measurement test bed with futek's loadcell and pressure display module.

Step response measurements were conducted at the end effector by applying a consistent step input to the regulator, aiming to assess both output delay and pressure loss along the transmission line. The regulator was configured with a 5V input, and the transmission line length was systematically varied from 1 m to 8 m in 1 m increments. The outcomes are shown in Table 3. Notably, the results consistently revealed delays of approximately 0.15 seconds, indicative of the time gap between the signal entering the regulator and the subsequent opening of the valve.

As the transmission line length increased, there was a noticeable extension in settling time, and the magnitude of the final convergence force exhibited a decline. Specifically, at a length of 1 m, a force of approximately 8.5 N was attained within 0.5 seconds. In contrast, at a length of 8 m, the force diminished to 5 N, and the settling time increased to 0.847 seconds.

**Open-loop force control of pressure display.** The force control mechanism of the pressure display module operates in an open-loop structure. To achieve the desired force output, a force-signal model was derived, taking into account the inherent hysteresis in pneumatic systems, and calibrated using a force sensor. Data on the correlation between the regulator signal and the force generated from the module were collected to establish a second-order force-signal model through quadratic regression analysis, as depicted in Fig 7a. The module was securely fixed with an acrylic block, and force was measured with the load and force sensor properly aligned. The force-signal model was formulated as a quadratic equation, as illustrated in Fig 7b. The equation exhibits a positive slope when the valve is open and a negative

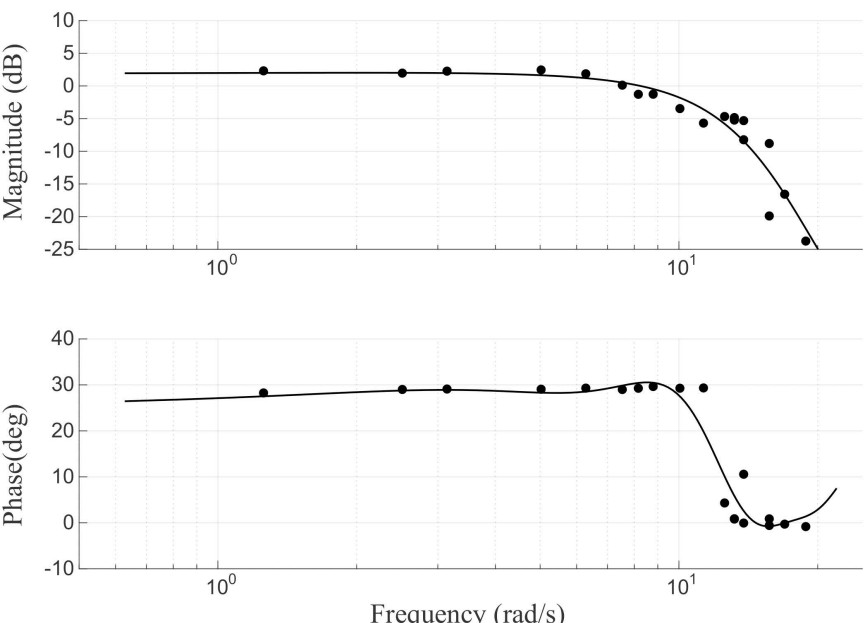

**Fig 6. Bode plot of the system in open-loop position control.** A sinusoidal pressure input with increasing frequencies (from 0.1 to 3 Hz) is commanded to the regulator, while the output position is measured at the pressure module.

**Table 3. Delay time of output according to system input.**

| Line Length (m) | Settling time (s) |
| --- | --- |
| 1 | 0.5012 |
| 2 | 0.5152 |
| 3 | 0.5281 |
| 4 | 0.6141 |
| 5 | 0.6990 |
| 6 | 0.7149 |
| 7 | 0.7966 |
| 8 | 0.8470 |

slope when the valve is closed. Signals were delivered in 1 N increments to assess the performance of the open-loop force control. The experimental results demonstrated smooth force delivery in the range of 1 N to 10 N, with an average absolute error of 0.046 N.

**fMRI-compatibility test.** Our experiments aimed to evaluate fMRI-compatibility by determining the impact of components and devices on MRI image quality. A crucial criterion for fMRI-compatibility is the absence of artifacts in scan results. Additionally, objects that exhibit significant magnetic attraction to the MRI bore pose a safety risk and are unsuitable for use. To assess safety, we tested objects for magnetic attraction using a neodymium magnet with 40 mm diameter, ensuring no harmful magnetic forces acted on the objects before MRI scanning.

To ascertain the fMRI-compatibility of the tested objects in the MRI environment, we used a water-filled phantom as a standard reference, placed on the RF head coil for signal reception. We then conducted Echo-Planar Imaging (EPI) sequence imaging with the test object and the phantom inside the MRI bore, comparing these images against

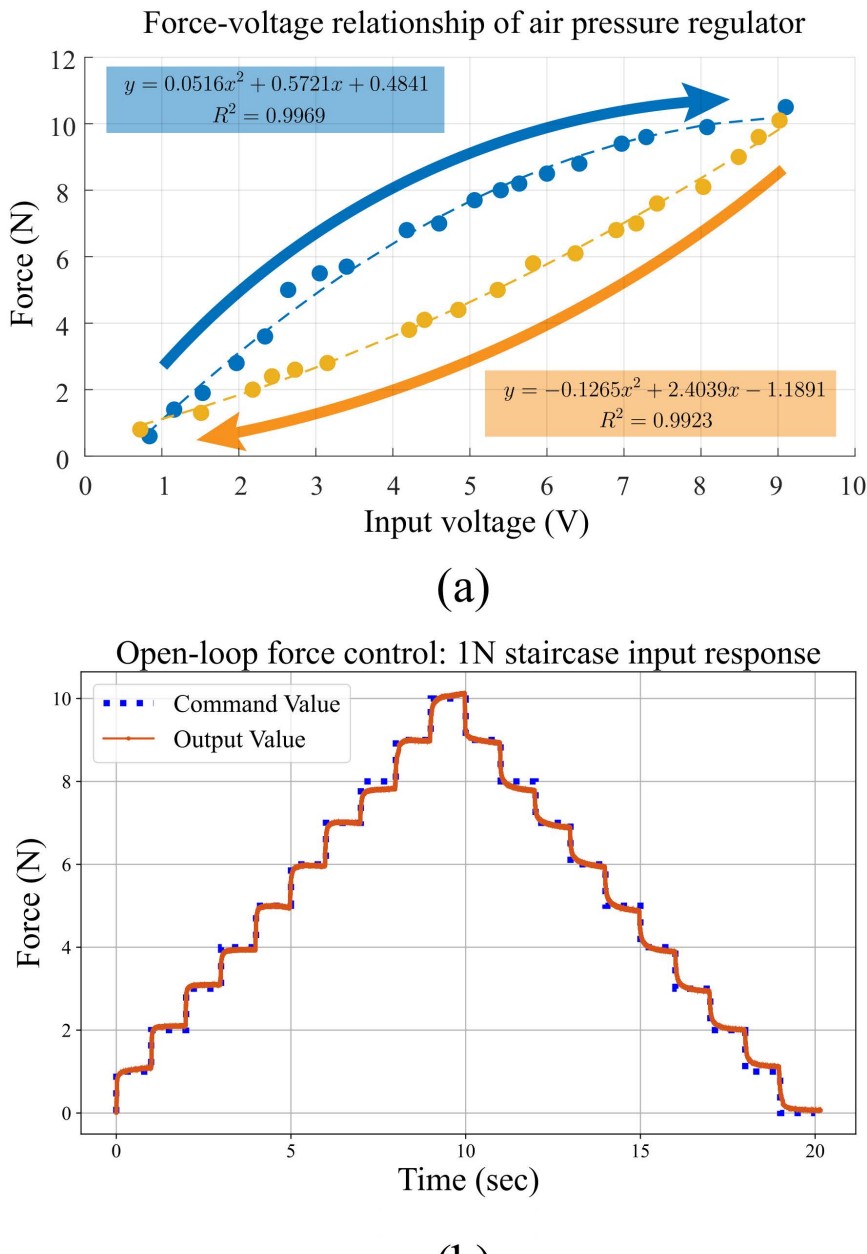

**Fig 7. (a) Signal-force graph for open loop force control.** (b) Open-loop force control results.

the baseline to assess fMRI-compatibility. The analysis utilized signal-to-noise ratio (SNR) and temporal signal-to-noise ratio (tSNR) for our proposed haptic device and its components and the data glove from Section V. Additionally, we performed statistical analysis on the tSNR for piezo disks, envisioned as texture displays for potential multimodal display development. The mean signal ($S_\mu$) and standard deviation of noise ($N_\sigma$) were calculated using the formula:

$$S_\mu = \frac{2}{Nt_{end}} \sum_{i=1}^{N} \sum_{t=1}^{t_{end}/2} \mu_i(t), \tag{1}$$

$$N_\sigma = \frac{2}{Nt_{end}} \sum_{i=1}^{N} \sum_{t=1}^{t_{end}/2} \sigma_i(t), \tag{2}$$

where N is the voxel count in the region of interest (ROI), t represents acquisition time, $\mu_i(t)$ is the mean signal within the signal area, and $\sigma_i(t)$ denotes the noise area's standard deviation. Given the EPI sequence's repetition time is 2 seconds and the presence of 3D voxel data over time, indices were derived by dividing by $2/Nt_{end}$.

The signal analysis focused on a circular area constituting 75% of the phantom's volume as the ROI, with four rectangular corners designated as the noise ROI. The tSNR calculation, following the methodology proposed by [26], accounts for temporal fluctuations under specific experimental conditions and is a standard metric in fMRI-compatible robotics research [11,27]. We calculated the tSNR for each voxel within a 30×30×12 voxel (10800 voxels) ROI at the center of the image data, providing a comprehensive assessment of fMRI-compatibility for the components and devices under investigation. In human test, 24×24×12 voxel (6912 voxels) was utilized as a ROI for tSNR calculation.

$$tSNR_i = 20 \log_{10} \frac{\text{Mean(voxel time series)}}{\text{Std(Detrend(voxel time series))}}, \tag{3}$$

The fMRI-compatibility tests were conducted using a Philips Achieva 3.0 T TX system equipped with a 32-channel RF head coil. Functional MR images were acquired using an Echo-Planar Imaging (EPI) sequence characterized by TR (repetition time) = 2000 ms, TE (echo time) = 30 ms, flip angle = 90°, matrix size = 80×80×35, and voxel size = 2.4×2.4×3.0 mm³, with a bandwidth of 1312 Hz/pixel.

The testing comprised three stages: compatibility assessments on components, on the haptic displays, and finally on human subjects. Initially, component tests were carried out to ascertain the viability of building an fMRI-compatible haptic display. Following the positive outcomes from these component tests, an fMRI-compatible haptic display was developed and subjected to further compatibility testing. Lastly, to understand if the presence and operation of the haptic displays influenced image quality, a test was performed involving human subjects.

## Experiment 1 – Compatibility test for components

The compatibility of each component is critical, as a single incompatible part can compromise the entire device. To systematically develop an fMRI-compatible device, we performed distance-based compatibility tests on the necessary components.

**Method.** For compatibility assessment, each component was placed at varying distances from the phantom, under the following conditions:

- Direct contact (0 cm)

- 15 cm away

- 30 cm away

- 45 cm away

The impact of each component on the signal's mean and noise standard deviation was compared against a baseline. The data glove (MANUS VR, Prime one), intended for VR content experimentation, underwent statistical analysis through its temporal signal-to-noise ratio (tSNR).

**Results.** The results, summarized in Fig 8, evaluate the compatibility of various components with fMRI environments using the Kruskal-Wallis test and Dunn's post hoc analysis. In the figure, baselines are indicated in red, incompatible results in blue, and compatible results in green. To account for MRI scanner drift and noise, nine components were divided into three groups for comparison against their respective baselines based on SNR and tSNR metrics. Components tested included copper wires, pneumatic joints, force sensing resistors (FSRs), optical fiber amplifiers, terminal blocks, acrylic plates, bolts (brass, SUS303, plastic), commercial pneumatic cylinders, and piezo disks.

SUS304, a ferromagnetic material, caused significant SNR drops and artifacts at close distances, while brass showed no artifacts beyond 15 cm. Optical fibers demonstrated compatibility across all distances, and pneumatic joints were compatible when placed beyond 30 cm. Terminal blocks (PCT-215) consistently caused SNR drops and artifacts, confirming their incompatibility. Commercial pneumatic cylinders showed artifacts within 30 cm but none at 45 cm, indicating partial compatibility. In contrast, plastic bolts and acrylic plates exhibited no artifacts, demonstrating full compatibility.

Piezo disks of 25.3 mm diameter, which can simulate touch using squeeze film levitation but require high voltages and frequencies, were assessed for their impact on tSNR under operational conditions. Piezo disks, tested at 45 cm, caused no significant impact on tSNR regardless of operational status, confirming compatibility at sufficient distances. These findings emphasize the importance of material properties and placement for ensuring fMRI-compatibility, with optical fibers, plastic and brass bolts, acrylic plates, and piezo disks at appropriate distances proving most suitable.

The data glove's fMRI-compatibility was statistically confirmed using tSNR, with tests showing no significant impact on image quality across experimental conditions at 45 cm in post hoc validation (p = 0.0797).

## Compatibility test for haptic display

With insights from Experiment 1, the developed haptic display underwent compatibility testing to ascertain if its operation adversely affected fMRI images.

**Method.** The testing setup placed the proposed haptic display on the MRI bed next to the phantom under several conditions:

- Baseline (absence of the device)

- Device at 0 cm from the phantom, not operating

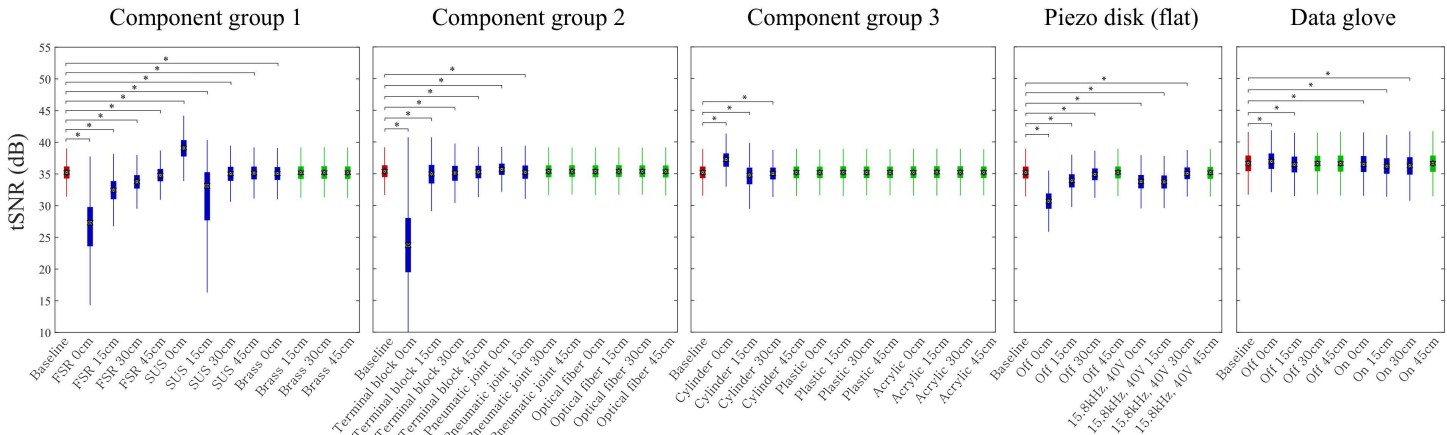

**Fig 8. Box plot of tSNR for fMRI-compatibility test about various components.** Baselines are shown in red, incompatible components are shown in blue, and compatible components are shown in green. (*) represents a significance level of 0.05 in Dunn's test, a post hoc test.

- Device at 0 cm from the phantom, operating

- Device at 67 cm from the phantom, not operating

- Device at 67 cm from the phantom, operating

Given the typical hand position from the RF coil when a subject lies in the MRI (67 cm), these conditions aimed to mimic real-world use. A t-test, along with the non-parametric Kruskal-Wallis test, was applied for statistical analysis against the baseline under each condition.

**Results.** The haptic display was determined to be fMRI-compatible as indicated in Table 4, with no observed artifacts or image distortion under any condition. Although the "power on" at 0 cm condition deviated from the null hypothesis, other settings demonstrated compatibility. For conditions involving the device at 67 cm, whether operating or not, a Kruskal-Wallis test verified no significant impact on tSNR (p = 0.9182) (Fig 9a).

### Experiment 3 – Compatibility testing with a human subject

This phase aimed to verify the fMRI-compatibility of the proposed haptic display with human subjects performing hand-grasping motions.

**Method.** A single male subject, 27 years old, right-handed, with no implants affecting fMRI, underwent three imaging sessions:

**Table 4. fMRI-compatibility test result of the proposed haptic display.**

| Condition | $S_\mu(N_\sigma)$ | SNR | tSNR | p-value |
|---|---|---|---|---|
| Baseline | 1192.76(0.60) | 2002.70 | 35.07 | |
| Haptic display off, 0 cm | 1195.92(0.62) | 1918.21 | 35.07 | 0.879 |
| Haptic display on, 0 cm | 1207.80(0.61) | 1967.95 | 34.52 | < 0.001 |
| Haptic display off, 67 cm | 1198.03(0.62) | 1952.63 | 35.08 | 0.918 |
| Haptic display on, 67 cm | 1192.76(0.61) | 1951.76 | 35.07 | |

Signal, noise and tSNR data were measured and averaged according to each ROI.

The p-value for 0 cm condition corresponds to the post-hoc test.

The p-value for 67 cm condition corresponds to the Kruskal-Wallis test.

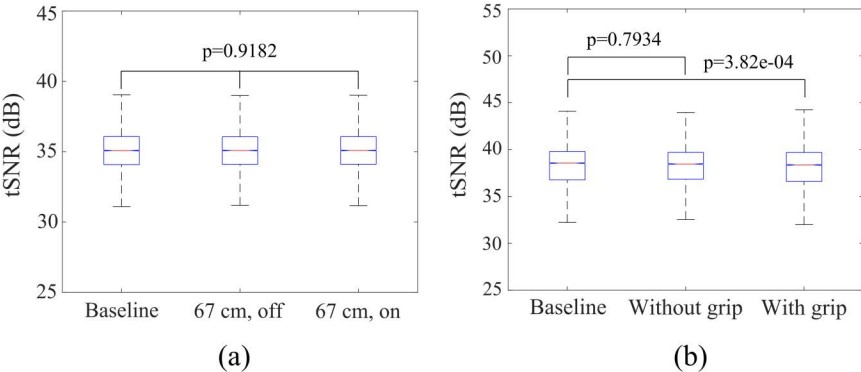

(a)　　　　　　　　(b)

**Fig 9. Box plot of tSNR for fMRI-compatibility test in different experimental conditions.** The box represents the interquartile range (IQR), showing the central 50% of the data. The whiskers extend to the smallest and largest values within 1.5 times the IQR from the lower and upper quartiles, respectively. (a) phantom tests with haptic display, and (b) human subjects tests with haptic display.

- Baseline (before system installation)

- With the system installed but not gripped

- Gripping the system

A two-sample t-test was conducted on the tSNR for each condition.

**Results.** T-test results regarding tSNR indicated no significant effect (p = 0.7934) for the haptic display's presence without grip (Fig 9b). The detailed quantitative metrics for each condition, including SNR and tSNR, are presented in Table 5. However, gripping the device significantly altered the results compared to the baseline (p = 3.82e-04), attributable to dynamic changes in the BOLD signal during gripping. These findings, alongside Experiment 2, confirm the proposed haptic display system's fMRI-compatibility.

## Experimental evaluation: Haptic display in VR with fMRI results

To investigate the effect of our proposed haptic display on the immersion of VR experiences, we developed VR content that simulates piccolo trumpet performance, as illustrated in Fig 10. This content was created using Unity and designed to be compatible with our multi-finger haptic display. The virtual reality trumpet playing system enabled subjects to view a monitor screen positioned above their head through a mirror projection system, as shown in Fig 1. Trumpet sounds were delivered via pneumatic headphones. Both the monitor display and pneumatic headphones were MRI-compatible. Considering the complexity of playing a trumpet, which involves both fingering and breath control, we simplified the task for participants unfamiliar with the instrument by fixing the breath input to a constant level. This modification allowed pitch changes to be determined solely by fingering, with cylinders to be pressed highlighted in red for ease of learning.

**Table 5. fMRI-compatibility test result of human subject.**

| Condition | $S_\mu(N_\sigma)$ | SNR | tSNR |
|---|---|---|---|
| Baseline | 1449.78(1.27) | 1137.13 | 37.84 |
| Without grip | 1396.85(1.24) | 1125.85 | 37.83 |
| With grip | 1450.00(1.26) | 1155.42 | 37.64 |

Signal, noise and tSNR data were measured and averaged according to each ROI.

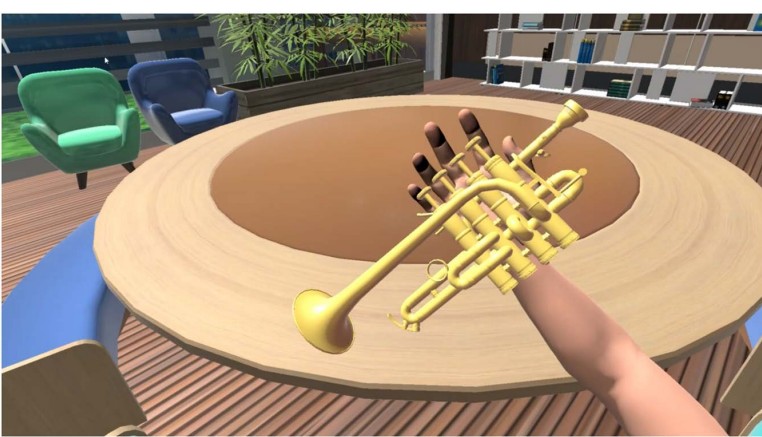

**Fig 10. VR content for playing piccolo trumpet.**

This experiment focused on playing the trumpet, a task that does not utilize texture display, aiming to quantify immersion enhancements attributed to haptic feedback. Unlike data gloves that merely track hand movements without providing sensory feedback, our haptic display intends to enhance immersion by introducing haptic feedback, measurable through the activation of sensory-related brain regions. We hypothesize that increased activation in these areas when using the haptic display, compared to the data glove, would indicate a higher level of immersion due to haptic rather than visual feedback alone. Integration of the haptic display and data glove with the VR content was achieved through TCP-IP communication, linking C++ code controlling the haptic display with Unity.

Additionally, to reinforce the link between haptic feedback and immersion, we compared two scenarios in which haptic feedback was continuously provided at the same intensity. We investigated temporal synchronization and desynchronization between visual and haptic feedback.

As shown in the experimental protocol in Fig 11, subjects repeated two sessions consisting of "Play" and "Rest" blocks in both experiments. Each condition of each experiment occurred in one session, and the order of the sessions was randomized for each subject. A 120-second interval between sessions ensured no carryover of activation from the first to the second session. A "Rest" block, lasting for 20 seconds, involved no active tasks. During the "Play" block, subjects engaged in trumpet playing through finger movements, receiving visual, auditory, and haptic feedback. The trumpet performance task was implemented using a structured format in which participants responded to visually cued button sequences. Random sets of buttons were highlighted in red and presented at regular 4-second intervals throughout the task. Each highlighted button configuration remained visible for 3 seconds, followed by a 1-second inter-stimulus interval

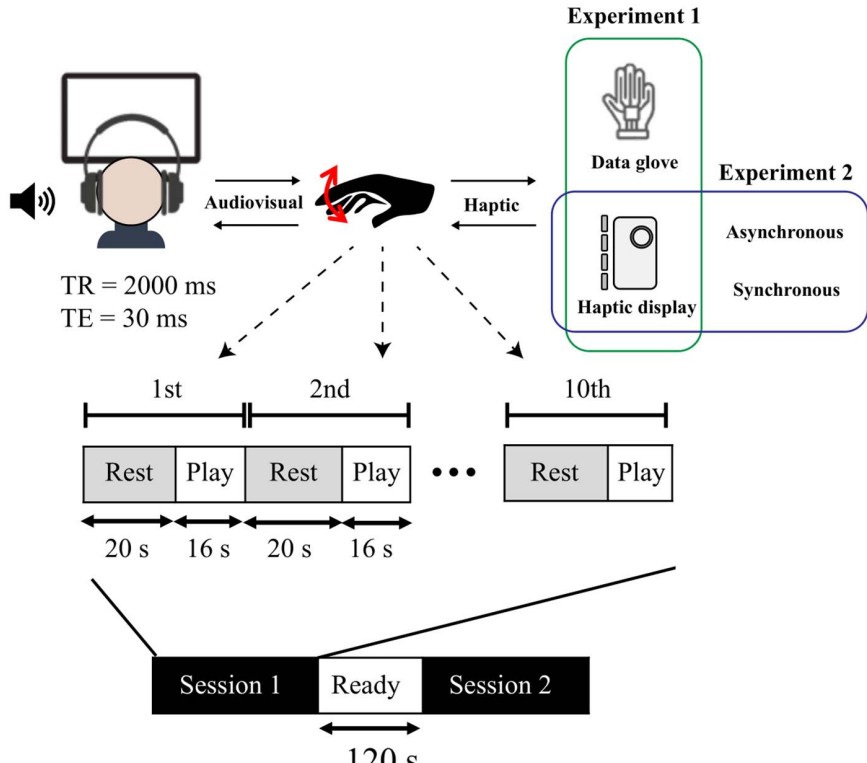

**Fig 11. fMRI experiment protocol.**

before the subsequent set appeared. Participants were instructed to press the corresponding buttons as they became highlighted on the display. Each session included ten cycles of Rest/Play blocks.

In the first experiment, to eliminate the movements associated with donning and doffing the data gloves, the protocol divided the experiment into sessions using the data gloves (WD) and the haptic display (WH). In the second experiment, the experimental protocol consists of a session in which haptic feedback is synchronized with audiovisual feedback (Synchronous) and a session in which haptic feedback is desynchronized (Asynchronous). In the asynchronous session, asynchronous was ensured by detecting the subject's button press and subsequently delivering haptic feedback with a 500 ms delay.

**Participants.** This study was approved by the Research Ethics Committee of Pohang University of Science and Technology (approval number: PIRB-2022-E015), and written informed consent was obtained from all participants. Human participants were recruited from November 1, 2022, to December 21, 2023.

For Experiment 1, a total of 13 healthy subjects (3 female; mean age = 23.23 ± 1.31 years) participated. For Experiment 2, a separate group of seven healthy participants (2 female; mean age = 24.57 ± 2.41 years) was recruited. For both experiments, none had a history of neurological disorders or impairments in vision, hearing, or motor sensation. The tasks were performed with the right hand.

**fMRI data acquisition.** Scanning was performed using a Philips Achieva 3.0 T TX system equipped with an 32-channel RF head coil. Functional MR images were acquired with an EPI sequence detailed as follows: TR (repetition time) = 2000 ms, TE (echo time) = 30 ms, flip angle = $90°$, matrix size $80{\times}80{\times}35$, voxel size $2.75{\times}2.75{\times}4.00$ mm$^3$. Anatomical reference was obtained through 3D T1 images. Functional images were preprocessed using SPM12 (Statistical Parametric Mapping) in MATLAB 2022b.

The preprocessing pipeline consisted of the following steps, performed in order: (1) realignment to correct for head motion, (2) spatial normalization to an East Asian brains ICBM space template, and (3) spatial smoothing with an 8-mm Full Width at Half Maximum (FWHM) Gaussian kernel.

**fMRI data analysis.** First-level statistical analyses were performed for each subject. The BOLD signal was modeled using the canonical hemodynamic response function in SPM12. A high-pass filter with a 128-second cutoff period was applied to the time series, and temporal autocorrelation was modeled using a first-order autoregressive model. Six head motion parameters generated by the realignment in SPM12 and a session-specific constant were included as confound regressors for each of the two imaging runs, resulting in a total of 14 confound regressors per subject.

For Experiment 1, the design matrix was constructed with four regressors of interest: 'With Data Glove' (WD) play, WD rest, 'With Haptics' (WH) play, and WH rest. The corresponding contrasts, which applied to these regressors of interest while setting all confound regressors to zero, were defined as: WD > Rest ([1−1 0 0]), WH > Rest ([0 0 1−1]), WH > WD ([−1 1 1− 1]), and its inverse, WD > WH ([1 − 1 −1 1]). For Experiment 2, the design matrix included four different regressors of interest: Synchronous play, Synchronous rest, Asynchronous play, and Asynchronous rest. Similarly, the contrasts applied only to these regressors and were defined as: Synchronous > Rest ([1−1 0 0]), Asynchronous > Rest ([0 0 1−1]), Synchronous > Asynchronous ([1 − 1 −1 1]), and Asynchronous > Synchronous ([−1 1 1− 1]).

For both experiments, the resulting contrast images from each subject were carried forward to a second-level, random-effects group analysis. A one-sample t-test model was used for each contrast, with statistical significance assessed using non-parametric permutation testing (5000 permutations) as implemented in the SnPM13 toolbox [28]. For cluster-wise statistical inference across the whole brain, a cluster-forming threshold of p > 0.001 (uncorrected) was applied.

## Results

No clusters survived the cluster-level FDR correction ($p < 0.05$) in either experiment. However, for exploratory purposes and to guide future works, we report the results identified using a cluster-forming threshold of $p < 0.001$ (uncorrected). For each contrast, the 4 largest clusters are reported for descriptive purposes (all $_{PFDR} > 0.05$).

## Experiment 1

The VR environment effectively rendered the trumpet-playing task. Activations in both WH and WD sessions included the contralateral primary motor cortex (M1), bilateral somatosensory cortices (S1, S2), supplementary motor area (SMA), and cingulate motor area (CMA), aligning with previous grip-related studies. The WH > WD contrast revealed more extensive brain activation in sessions involving the haptic display, notably in the bilateral postcentral gyrus as demonstrated in Fig 12. No areas showed increased activation with the WD > WH. Analysis of head movement indicated minimal movement artifacts. Subjects reported no discomfort while using either the haptic device or the data glove. The anatomical locations of the activation clusters were identified using the atlases listed in Table 6.

## Experiment 2

In both the Synchronous > Rest and Asynchronous > Rest conditions, activation was evident in the left primary somatosensory cortex and the primary motor cortex, corresponding to the contralateral side of the right hand where the haptic device was utilized.

In the analysis of Synchronous > Asynchronous, As illustrated in Fig 13, activation was observed in the left cerebellum and lingual gyrus (LING). Additionally, a large cluster was identified with its peak in the left white matter (MNI: −40, −12, 26), which extended inferiorly to encompass a portion of the left insula. In the right hemisphere, activations were found in the fusiform gyrus and the parahippocampal gyrus (PHG).

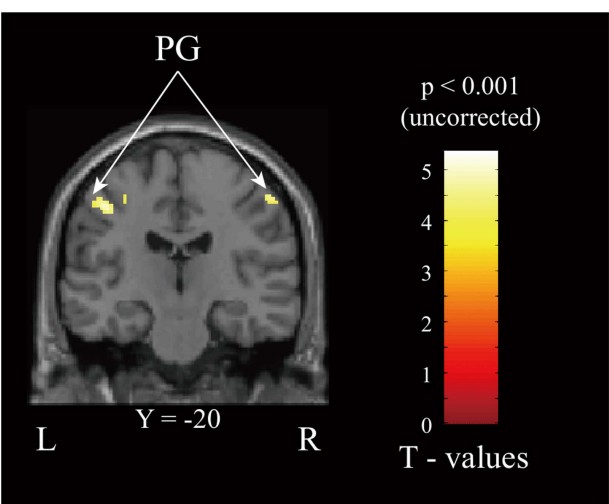

**Fig 12. Neurological view corresponding to group fMRI results showing dominant activation regions in the WH > WD contrast (PG – Postcentral gyrus, blue $p_{FDR}$ > 0.05).**

**Table 6. Brain regions showing activation for the WH > WD contrast. Clusters were identified using a cluster-forming threshold of $p < 0.001$ (uncorrected). The 4 largest clusters are shown for exploratory purposes (all $_{pFDR}$ > 0.05). Anatomical locations were identified using the Automated Anatomical Labeling (AAL) atlas.**

| Anatomical Region | Cluster size | $p_{unc}$ | T | x | y | z |
|---|---|---|---|---|---|---|
| L. Postcentral Gyrus | 102 | 0.0002 | 4.98 | −42 | −20 | 46 |
| R. Postcentral Gyrus | 25 | 0.0004 | 4.54 | 54 | −20 | 54 |
| L. White Matter | 16 | 0.0008 | 4.84 | −24 | −4 | 22 |
| R. White Matter | 15 | 0.0002 | 4.93 | 20 | 10 | 20 |

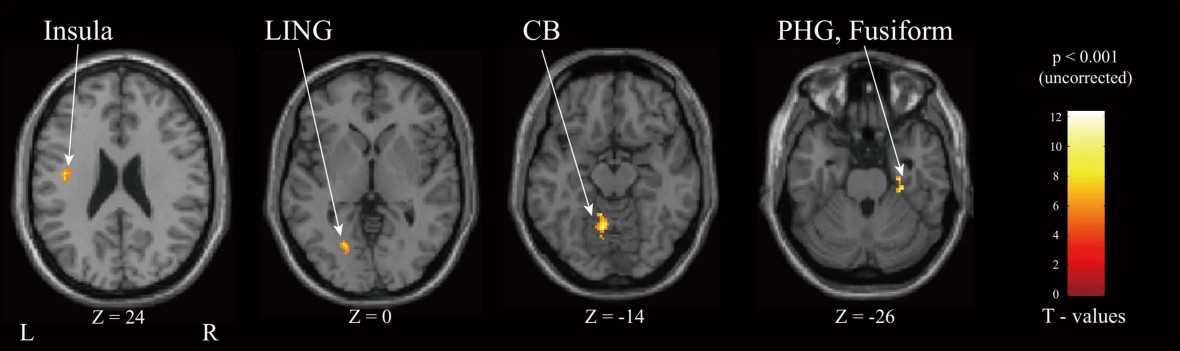

**Fig 13. Neurological view corresponding to group fMRI results showing dominant activation regions in the Synchronous > Asynchronous contrast (LING – lingual gyrus, CB – Cerebellum, PHG – Parahippocampal Gyrus, all $p_{FDR}$ > 0.05).**

No areas showed increased activation with the Async > Sync. Analysis of head movement indicated minimal movement artifacts. Subjects reported no discomfort while using either the haptic device. The anatomical locations of the activation clusters were identified using the atlases listed in Table 7.

## Discussion

This paper demonstrates a novel handheld fMRI-compatible multi-finger haptic display. In this section, we evaluate the performance of the proposed haptic display and describe experimental results that quantify the immersiveness achieved using the proposed display in VR environments.

### Haptic performance

To realize this system, we first engineered an improved pneumatic cylinder for our multi-finger haptic display, focusing significantly on compactness and sealing to ensure fMRI-compatibility. Subsequently, we developed VR content to evaluate the reality and immersion provided by the proposed display. The test results from the open-loop force control for our proposed pneumatic cylinder, featuring an acrylic multi-layer construction, indicate it can provide forces ranging from 1–10 N with a resolution of 1 N. The cylinder's stroke extends to 13 mm and it operates up to 1.5 Hz. The force range of the actuator falls within the typical spectrum of haptic forces experienced by users, ensuring safety due to its limited stroke.

**Table 7. Brain regions showing activation for the Sync > Async contrast Clusters were identified using a cluster-forming threshold of $p < 0.001$ (uncorrected). The 4 largest clusters are shown for exploratory purposes (all $_{pFDR}$ > 0.05). Anatomical locations were identified using the Automated Anatomical Labeling (AAL) atlas.**

| Anatomical Region | Cluster size | $p_{unc}$ | T | x | y | z |
|---|---|---|---|---|---|---|
| L. White Matter | 46 | 0.0078 | 12.32 | −40 | −12 | 26 |
| L. Cerebellum | 47 | 0.0078 | 11.30 | −8 | −58 | −14 |
| | | 0.0078 | 8.56 | −10 | −46 | −12 |
| R. PHG | 25 | 0.0078 | 10.10 | 28 | −18 | −26 |
| R. Fusiform | | 0.0078 | 10.09 | 30 | −26 | −26 |
| L. LING | 28 | 0.0078 | 7.63 | −22 | −66 | 0 |

PHG – Parahippocampal Gyrus.

LING – lingual gyrus.

Our haptic device, which provides force feedback to four fingers, adds a sense of realism to various scenarios. For example, users can directly feel with their fingertips the pressure of pressing a virtual button, the tension of drawing a bowstring, and the difference in stiffness between a soft sponge and a hard brick.

The use of accurate sensors that generate or are operated by magnetic fields poses challenges in fMRI settings, making it difficult to incorporate compact and precise sensors within this environment. Therefore, we opted for an open-loop control to render pressure, independent of sensor feedback. This control limitation, attributed to the pneumatic actuation method and the extended transmission line, introduces a time delay and reduces the actuation frequency. While this limits tasks requiring rapid finger movements, such as fast-paced piano playing, it remains suitable for most simple daily tasks. Separate calibration of each cylinder is necessary due to individual performance variations stemming from minimal tolerance differences and the non-linearity of pneumatic pressure.

Experiments have confirmed its full fMRI-compatibility, with no incidents or accidents reported over 30 hours of fMRI use. The device did not produce spatial and temporal noise or artifacts, supported by the tSNR values, due to its composition solely of paramagnetic bolts, nuts, and non-metallic materials.

## Experiments and fMRI results

The fMRI experiments with VR trumpet performance highlight the effectiveness of our multi-finger haptic device in quantifying immersion. Increases in brain activity were observed when haptic feedback was present and synchronized with audiovisual cues. Increases occurred in areas related to sensation, such as coordination of movement, and areas related to cognition, such as attention and sustained concentration.

The comparison between the presence and absence of haptic feedback (WH > WD) revealed stronger activation in the contralateral postcentral gyrus. This is consistent with its established role in integrating finger-specific motor and tactile information [29] and in object manipulation [30].

Furthermore, the Synchronous > Asynchronous analysis highlighted a network of regions involved in binding multisensory information into a coherent whole. Activation in the lingual gyrus (LING) aligns with its function in processing complex visual information [31]. The engagement of the fusiform gyrus is particularly noteworthy, as this region is associated with processing semantic congruency between haptic objects and their related sounds [32]. Moreover, the parahippocampal gyrus (PHG) activation is consistent with its specific role in encoding kinematic parameters, such as speed and direction, during visuomotor tasks. Together, this pattern suggests that synchronous feedback facilitates the integration of visual, haptic, and contextual spatial information into a meaningful percept.

The cerebellum's involvement can be attributed to its role in performance monitoring and temporal prediction [33,34], both of which are essential for synchronizing actions with sensory feedback in timed-response tasks. Crucially, the insula has been implicated in higher-order cognitive control and attention [35]. Several reports have noted increased insula activation as subjects became more aware of being in control of an action [36,37]. Also, its known involvement in stimulus-driven attention and self-awareness processes has been directly linked to the sense of presence [38].

These findings provide preliminary evidence that haptic feedback increases sensory fidelity, thus improving immersion in VR tasks like trumpet performance.

Our multi-finger haptic device is adaptable for a range of tasks with multiple degrees of freedom. While the present study focused on a trumpet performance task, our experimental paradigm is broadly applicable to any rich multisensory scenario where haptic feedback from physical interaction occurs in concert with congruent audiovisual cues. This includes applications such as studying fine force interactions during keyboard typing, or providing tactile feedback for virtual object surface recognition, where the interplay between seeing, hearing, and feeling is critical.

## Limitations

We acknowledge that the relatively small sample sizes (n = 13 for group A and n = 7 for group B) and using uncorrected p for analysis are a significant limitation of this study. Furthermore, a primary limitation of the present study is the absence

of subjective behavioral data, such as participant ratings on their sense of immersion. While our fMRI results suggest potential neural correlates, future research is required to strengthen these findings by demonstrating a direct correlation between these brain activations and participants' self-reported experiences. Such studies will be essential to elucidate the precise mechanisms by which these neural patterns contribute to an enhanced sense of immersion and to comprehensively map the relationship between objective neural markers and the subjective feeling of immersion.

It should also be noted that the trumpet task resembles a button-press paradigm. However, our device distinguishes itself by providing realistic pneumatic resistance rather than the binary click of standard input devices. We recognize that this specific task did not fully exploit the device's variable force capabilities, necessitating future research with more dynamic tasks.

Despite these limitations, this study serves as a crucial pilot investigation, successfully generating hypotheses for future research. Our haptic experiments provided valuable preliminary data, offering detailed insights into quantifying immersion.

In addition to these methodological expansions, our future technical roadmap involves significant hardware advancements. We will incorporate position control and express texture using the piezo disk mentioned in the component compatibility test. The question of how immersive VR experiences will be enhanced by the fMRI-compatible multi-finger haptic display combined with texture feedback remains open, suggesting the possibility of extending beyond multi-finger to a multi-modal domain.

## Conclusion

In our study, we introduced an innovative fMRI-compatible handheld multi-finger pneumatic haptic display, designed to integrate with VR content. Through rigorous testing within a 3T MRI environment, we meticulously evaluated the MRI compatibility of our haptic display's components, leading to the development of a system that accurately produces tactile feedback across four fingers, with force ranges from 1N to 10N. This capability is complemented by the system's provision of comprehensive sensory experiences, encompassing visual, auditory, and tactile feedback, which were specifically tailored for VR interactions involving a piccolo trumpet.

As a pilot investigation, our empirical study provided preliminary evidence regarding the potential influence of tactile feedback on immersion within VR environments. Although the sample size was limited, the observed patterns suggest that the incorporation of haptic feedback may correspond with modulated activation in brain regions associated with immersion. These results demonstrate the feasibility of using our haptic display to objectively assess the VR experience.

The implications of our findings extend beyond the technical achievements, opening new avenues for research and application in fields where immersive training and education are critical. As we continue to refine our technology and explore broader applications, the potential for further enhancing the realism and effectiveness of VR experiences remains vast, promising unprecedented opportunities for advancement in digital interaction.

## Author contributions

**Conceptualization:** Joonsub Byun, Joonseon Hwang, Keehoon Kim.

**Data curation:** Joonsub Byun.

**Formal analysis:** Joonsub Byun, Yong-An Chung, Hyeonseok Jeong.

**Funding acquisition:** Keehoon Kim.

**Investigation:** Joonsub Byun, Joonseon Hwang.

**Methodology:** Joonsub Byun, Keehoon Kim.

**Project administration:** Keehoon Kim.

**Resources:** Jooyeon Kim.

**Software:** Joonsub Byun, Joonseon Hwang.

**Supervision:** Keehoon Kim.

**Validation:** Joonsub Byun, Joonseon Hwang.

**Visualization:** Joonsub Byun.

**Writing – original draft:** Joonsub Byun.

**Writing – review & editing:** Joonsub Byun, Keehoon Kim.

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
