## [Decision Letter · Decision Letter 0]

5 Jun 2025

Dear Dr. Kim,

Thank you for submitting your manuscript to PLOS ONE. After careful consideration, we feel that it has merit but does not fully meet PLOS ONE’s publication criteria as it currently stands. Therefore, we invite you to submit a revised version of the manuscript that addresses the points raised during the review process.

**ACADEMIC EDITOR:**

Before I consider accepting your manuscript, please comprehensively address the comments and concerns from Reviewer #1, Reviewer #2, as well as my own comments.

To resolve Reviewer #1's Comment #4, I would like you to cautiously or multidimensionally discuss the functionality of the identified brain regions. Please provide additional literature support that justifies your experimental paradigm and clearly outlines potential error ranges, emphasizing the consistency of BOLD signal changes.

**Minor revisions:**

**Page 3:** In Table 1, please replace "O" and "X" with clear, universally understandable terms such as "yes/no" or "with/without," to avoid ambiguity for readers from diverse linguistic and cultural backgrounds.In Table 1, please replace "O" and "X" with clear, universally understandable terms such as "yes/no" or "with/without," to avoid ambiguity for readers from diverse linguistic and cultural backgrounds.**Page 10:** Please remove the numbering "0.1" from the subsection titled "Experimental Evaluation: Haptic Display in VR with fMRI Results" under the Results section.Please remove the numbering "0.1" from the subsection titled "Experimental Evaluation: Haptic Display in VR with fMRI Results" under the Results section.**Figure 7 (b):** Adjust labels so they do not overlap or obscure the plotted curves.Adjust labels so they do not overlap or obscure the plotted curves.**Figure 9:** Clearly explain the meaning of bins and the error bars (whiskers) in the figure legend or descriptive text accompanying the figure.Clearly explain the meaning of bins and the error bars (whiskers) in the figure legend or descriptive text accompanying the figure.

We look forward to receiving your revised manuscript.

Kind regards,

Ziyu Qi

Academic Editor

PLOS ONE

Journal Requirements:

4. Thank you for stating in your Funding Statement: [This work was supported in part by Institute of Information & Communications Technology Planning & Evaluation (IITP) grant funded by the Korea government(MSIT) (No.2021-0-00986, Development of Interaction Technology to Maximize Realization of Virtual Reality Contents using Multimodal Sensory Interface), in part by the National Research Foundation of Korea (NRF) grant funded by the Korea government (MSIT) (No.2022M3C1A3081359), in part by the Pilot Project for Commercialization of Police Technology Public Research through the Commercializations Promotion Agency for R&D Outcomes funded by the National Police Agency and the Ministry of Science and ICT (No. 1711174175).].

Before we proceed with your manuscript, please address the following prompts.

Additional Editor Comments:

Before I consider accepting your manuscript, please comprehensively address the comments and concerns from Reviewer #1, Reviewer #2, as well as my own remarks.

To resolve Reviewer #1's Comment #4, I would like you to cautiously or multidimensionally discuss the functionality of the identified brain regions. Please provide additional literature support that justifies your experimental paradigm and clearly outlines potential error ranges, emphasizing the consistency of BOLD signal changes.

Minor revisions:

Page 3: In Table 1, please replace "O" and "X" with clear, universally understandable terms such as "yes/no" or "with/without," to avoid ambiguity for readers from diverse linguistic and cultural backgrounds.

Page 10: Please remove the numbering "0.1" from the subsection titled "Experimental Evaluation: Haptic Display in VR with fMRI Results" under the Results section.

Figure 7 (b): Adjust labels so they do not overlap or obscure the plotted curves.

Figure 9: Clearly explain the meaning of bins and the error bars (whiskers) in the figure legend or descriptive text accompanying the figure.

Reviewers' comments:

Reviewer's Responses to Questions

**Comments to the Author**

1. Is the manuscript technically sound, and do the data support the conclusions?

Reviewer #1: Partly

Reviewer #2: Partly

2. Has the statistical analysis been performed appropriately and rigorously?

Reviewer #1: No

Reviewer #2: N/A

3. Have the authors made all data underlying the findings in their manuscript fully available?

Reviewer #1: No

Reviewer #2: Yes

4. Is the manuscript presented in an intelligible fashion and written in standard English?

Reviewer #1: Yes

Reviewer #2: Yes

Reviewer #1: Thank you for submitting this manuscript on the development of an fMRI-compatible multi-finger haptic feedback device. The study integrates engineering design with functional neuroimaging to explore the concept of “immersion” in virtual reality (VR), which is technically innovative and has potential value in the intersection of VR and neuroscience. However, there are several key issues that need to be addressed before the manuscript can be considered for publication:

1. The theoretical link between “immersion” and the experimental paradigm is unclear:

The manuscript does not sufficiently discuss the dimensions that contribute to immersive experiences. Most cited studies focus on tactile feedback, but neglect other widely recognized components in VR research such as visual, auditory, motor, and bodily presence factors.

The assumption that “tactile feedback = immersion” is theoretically unsubstantiated. Please provide a more comprehensive review of the literature and justify why tactile feedback alone can represent or significantly influence immersion. Why is touch prioritized as the dominant modality in this context?

fMRI captures BOLD signal changes, but the link between these neural signals and subjective immersion is not well established. How can you ensure that the observed brain activations reflect “immersion” rather than attention, task load, or movement-related factors?

2. Lack of direct evidence that tactile feedback enhances immersion:

The study does not include behavioral data such as questionnaires or ratings from participants about their sense of immersion. This is a major weakness.

Please consider adding subjective measures or explain why they were not collected. Also discuss whether brain activity alone can be used as a reliable sign of immersion.

3. Concerns about statistical methods:

The fMRI analysis uses an uncorrected threshold (p < 0.005). In addition, the first brain image figure uses a cluster size threshold of 20 voxels, while the second uses 30 voxels. The reason for this difference is not explained.

Were these thresholds chosen using data-driven methods? Or were they selected without clear rules?

Using different and uncorrected thresholds increases the chance of false positives and creates concern about flexibility in analysis. Also, there is no correction for multiple comparisons (such as FWE or FDR), and no corrected p-values are reported. This does not meet current standards for fMRI analysis (e.g., Eklund et al., 2016, PNAS).

The sample size is small (13 and 7 participants), so results should be interpreted with caution.

4. Interpretation of fMRI results may be overstated:

The activation of regions such as the dlPFC and FEF is more commonly associated with attention and task control, not immersion per se. The manuscript should offer a more careful discussion and avoid over-interpreting these activations.

The synchronous vs. asynchronous task design may introduce confounding factors, such as violation of temporal expectation, which could activate regions like the SMG due to error awareness or attentional shift—not necessarily reduced immersion.

5. Additional suggestions:

Please discuss whether the proposed experimental paradigm is generalizable to other types of VR tasks or user interactions.

If possible, provide examples or preliminary data showing the applicability of this haptic device in different immersive scenarios to support its broader relevance.

Figures are clear, the manuscript is well-structured, and the overall language is fluent. However, some technical sentences are overly long and may benefit from simplification for better readability.

Reviewer #2: This study introduces and evaluates an fMRI-compatible, handheld multi-finger haptic display designed to enhance and measure immersion in virtual reality through tactile feedback. The article is interesting and relevant to the field. However, certain aspects would benefit from further clarification. While the multi-finger haptic display is thoroughly described, the implementation of the VR environment is discussed in less detail.

Regarding my understanding of the manuscript, the authors refer to the use of VR content created in Unity and describe visual feedback being delivered to participants during fMRI scanning. However, it does not specify the hardware to present visual stimuli inside the MRI scanner. Given the magnetic field of an MRI, and based on Figure 1 provided in the manuscript, is using standard VR headsets feasible in this setup? Or is it a mirror projection? Also, is Fig. 1 part of the experiment setup? If affirmative, I recommend adding it to the methodology section.

Additionally, the manuscript does not specify the method used for auditory feedback delivery, which is essential for assessing immersion and may be affected by the ambient noise produced by the fMRI scanner.

Regarding the VR-based trumpet activity, could the authors clarify the interaction mechanics? Was the task open-ended, allowing users to play freely, or did it follow a structured format, such as guiding users through specific sequences, like rhythm games like Guitar Hero?

Another suggestion for improving the manuscript is to present the underlying neural mechanisms or brain regions the authors expect to observe during the experimental trials. For instance, while the cerebellum is discussed later in the results, it is not mentioned in the introduction. Including such expectations earlier would help contextualize the study’s goals and hypotheses.

**Do you want your identity to be public for this peer review?** For information about this choice, including consent withdrawal, please see our For information about this choice, including consent withdrawal, please see our Privacy Policy .

Reviewer #1: No

Reviewer #2: No

While revising your submission, please upload your figure files to the Preflight Analysis and Conversion Engine (PACE) digital diagnostic tool, https://pacev2.apexcovantage.com/ . PACE helps ensure that figures meet PLOS requirements. To use PACE, you must first register as a user. Registration is free. Then, login and navigate to the UPLOAD tab, where you will find detailed instructions on how to use the tool. If you encounter any issues or have any questions when using PACE, please email PLOS at . PACE helps ensure that figures meet PLOS requirements. To use PACE, you must first register as a user. Registration is free. Then, login and navigate to the UPLOAD tab, where you will find detailed instructions on how to use the tool. If you encounter any issues or have any questions when using PACE, please email PLOS at figures@plos.org . Please note that Supporting Information files do not need this step.. Please note that Supporting Information files do not need this step.

---

## [Author Response · Author response to Decision Letter 1]

15 Jul 2025

Please refer to the "Response to Reviewers" for our replies. We are grateful for their valuable feedback.

---

## [Decision Letter · Decision Letter 1]

13 Sep 2025

Dear Dr. Kim,

Thank you for submitting your manuscript to PLOS ONE. After careful consideration, we feel that it has merit but does not fully meet PLOS ONE’s publication criteria as it currently stands. Therefore, we invite you to submit a revised version of the manuscript that addresses the points raised during the review process.

ACADEMIC EDITOR: Please rerun some analysis to clarify doubts of the reviewer. The results must be backed up with support from literature in the Discussion section. Please avoid any overstated conclusions rather report facts and limitations if any.

We look forward to receiving your revised manuscript.

Kind regards,

Usman Ghafoor

Academic Editor

PLOS ONE

Journal Requirements:

Reviewers' comments:

Reviewer's Responses to Questions

Reviewer #1: All comments have been addressed

Reviewer #3: (No Response)

2. Is the manuscript technically sound, and do the data support the conclusions?

Reviewer #1: Yes

Reviewer #3: Partly

3. Has the statistical analysis been performed appropriately and rigorously?

Reviewer #1: Yes

Reviewer #3: No

4. Have the authors made all data underlying the findings in their manuscript fully available?

Reviewer #1: Yes

Reviewer #3: Yes

5. Is the manuscript presented in an intelligible fashion and written in standard English?

Reviewer #1: Yes

Reviewer #3: Yes

Reviewer #1: thank you for your detailed revisions and clarifications. From the perspective of fMRI methodology and interpretation, the current version is clear and well addressed. I only suggest that the exploratory nature of the study be more explicitly emphasized in the manuscript, as this is a crucial point for appropriately framing the findings.

Reviewer #3: For clarification I did not initially review the paper during the first round.

This is an interesting study introducing a new piece of technology for virtual reality environments that is MRI safe, and I can very much see the importance of this work. I have no issues with the first half of the paper regarding the product and its testing. I do, however, have concerns regarding the fMRI experiments done in the second half of the paper, especially the over interpretation of results that I feel need to be addressed before I can recommend publication.

Major comments:

Significance level for correcting fMRI results:

The current manuscript uses questionable fMRI analysis methods that does not meet current fMRI analysis standards. This was raised by a previous reviewer, and I don’t believe the authors have adequately addressed the concerns by putting this just as a limitation. I understand the limitations of sample size; however, this is not an adequate reason not to conduct fMRI analysis to modern analysis standards.

Please redo the analysis with an uncorrected threshold of p < 0.001 (which is a standard uncorrected threshold measure). Then do an adequate correction for multiple comparisons (ideally permutation testing) for all the contrasts. If the authors achieve no significant results, for the contrast WH > WD in experiment one, I could maybe see an argument to present the uncorrected findings so long as there isn’t speculation of neural mechanisms, but rather a discussion on the importance of haptic feedback.

Over interpreting of fMRI results:

The current manuscript overstates fMRI findings and makes claims that are not supported by the analysis methods. Again, this was raised by a previous reviewer, and I do not feel the authors in the updated manuscript have adequately addressed this. Given the small sample size and the questionable fMRI statistical methods there is a high probability that findings are not robust and as such any talk of psychological/neural mechanisms to explain these findings is not currently appropriate. This may change once a more rigours analysis has been done. If it doesn’t then a discussion on why results were not found is important instead of on possible mechanisms seems more appropriate.

Presentation and explanation of fMRI data:

More detail in the main manuscript is needed regarding fMRI modelling and results. At the pre-processing stage, please state exactly which steps were conducted and in what order.

At the first level models please put how the BOLD signal was modelled (i.e. which HRF, any derivatives etc) and if any filtering of the time series was conducted. Please provide details of confound regressors used in the model, such as head motion, and state how many regressors were used at this stage. Please also state how the temporal autocorrelation was modelled.

For the second level models, it needs to be clearly stated whether this was a whole brain, or a ROI based analysis. Please present the t statistic, the contrast estimates, the corresponding p-value, the x, y, z co-ordinates along with the size for each significant finding. Please also state which atlas was used to determine location of regions. Ideally this should be done in a table.

Minor comments:

Limitations about behavioural data.

The justification for not using behavioural measures does not seem to be a valid objection. It is also contradictory as the authors argue about influencing results but then recommend using behavioural work in future research. Please remove the justification part of not using behavioural data. Then just put that a primary limitation of the present study does not use behavioural data, and that future work should use behavioural data.

Limitations should have its own subheading.

In Table 4. fMRI-compatibility test result of the proposed haptic display

Please present p value for Haptic display on, 0 cm. Also, please indicate which p value the test applies (is it the post hoc or the Kruskal-Wallis)

Do you want your identity to be public for this peer review? For information about this choice, including consent withdrawal, please see our Privacy Policy .

Reviewer #1: No

Reviewer #3: No

While revising your submission, please upload your figure files to the Preflight Analysis and Conversion Engine (PACE) digital diagnostic tool, https://pacev2.apexcovantage.com/ . PACE helps ensure that figures meet PLOS requirements. To use PACE, you must first register as a user. Registration is free. Then, login and navigate to the UPLOAD tab, where you will find detailed instructions on how to use the tool. If you encounter any issues or have any questions when using PACE, please email PLOS at . PACE helps ensure that figures meet PLOS requirements. To use PACE, you must first register as a user. Registration is free. Then, login and navigate to the UPLOAD tab, where you will find detailed instructions on how to use the tool. If you encounter any issues or have any questions when using PACE, please email PLOS at figures@plos.org . Please note that Supporting Information files do not need this step.. Please note that Supporting Information files do not need this step.

---

## [Author Response · Author response to Decision Letter 2]

13 Oct 2025

Thank you for your reviews and for the opportunity to revise our manuscript. Please refer to the attached Response to Reviewers file.

---

## [Decision Letter · Decision Letter 2]

29 Oct 2025

Dear Dr. Kim,

Thank you for submitting your manuscript to PLOS ONE. After careful consideration, we feel that it has merit but does not fully meet PLOS ONE’s publication criteria as it currently stands. Therefore, we invite you to submit a revised version of the manuscript that addresses the points raised during the review process.

We look forward to receiving your revised manuscript.

Kind regards,

Usman Ghafoor

Academic Editor

PLOS ONE

Journal Requirements:

**Additional Editor Comments:**

Please clarify issues raised by the reviewer.

Reviewers' comments:

Reviewer's Responses to Questions

**Comments to the Author**

Reviewer #3: All comments have been addressed

2. Is the manuscript technically sound, and do the data support the conclusions?

Reviewer #3: Yes

3. Has the statistical analysis been performed appropriately and rigorously?

Reviewer #3: I Don't Know

4. Have the authors made all data underlying the findings in their manuscript fully available?

Reviewer #3: Yes

5. Is the manuscript presented in an intelligible fashion and written in standard English?

Reviewer #3: Yes

Reviewer #3: Thank you to the authors for re-doing the analysis and addressing my comments. However, before I can endorse publication, I am confused regarding what the authors have done regarding second level fMRI analysis andI am hoping they can clarify. It seems that the authors have conducted correction for multiple comparisons by permutation testing, but then ignored the permutation test’s cluster extent threshold, instead using an arbitrary cluster extent of k > 20. If this is the case, their correction for multiple comparisons again does not meet modern neuroimaging standards. “Significant” clusters in this context will be those who meet the cluster extent threshold of corrected (FWE or FDR) p <0.05 and the cluster defining threshold of p <0.001 (uncorrected). However, if I am misunderstanding what the authors have done, then I apologise.

Can the authors clarify:

1) If they used cluster-based correction with a cluster forming threshold (p<0.001 uncorrected) and a cluster extent threshold (p corrected <0.05).

2) The cluster extent threshold is not defined by an arbitrary k, but on a corrected p value (p FWE/FDR <0.05) derived from the permutation tests.

If the authors have based the cluster extent threshold on an arbitrary level (k > 20) rather than on a corrected p value from the permutation tests, then:

1) The authors must define the cluster extent threshold as a corrected (FWE/FDR) p <0.05 derived from the permutation tests.

2) Only clusters that survive this level of multiple comparisons correction should be reported.

3) I would encourage the authors to read the cluster failure paper (Eklud et al., 2016) which persuasively shows why using arbitrary cluster extents does not control the type I error rate.

**Do you want your identity to be public for this peer review?** For information about this choice, including consent withdrawal, please see our For information about this choice, including consent withdrawal, please see our Privacy Policy .

Reviewer #3: No

---

## [Author Response · Author response to Decision Letter 3]

7 Nov 2025

Thank you for your valuable feedback. Please find attached our Response to the Reviewers.

---

## [Decision Letter · Decision Letter 3]

29 Jan 2026

Dear Dr. Kim,

Thank you for submitting your manuscript to PLOS ONE. After careful consideration, we feel that it has merit but does not fully meet PLOS ONE’s publication criteria as it currently stands. Therefore, we invite you to submit a revised version of the manuscript that addresses the points raised during the review process.

We look forward to receiving your revised manuscript.

Kind regards,

Usman Ghafoor

Academic Editor

PLOS One

Journal Requirements:

Additional Editor Comments:

The authors need to address some minor comments before the final decision.

Reviewer's Responses to Questions

**Comments to the Author**

Reviewer #4: (No Response)

2. Is the manuscript technically sound, and do the data support the conclusions?

Reviewer #4: Partly

3. Has the statistical analysis been performed appropriately and rigorously?

Reviewer #4: No

4. Have the authors made all data underlying the findings in their manuscript fully available?

Reviewer #4: (No Response)

5. Is the manuscript presented in an intelligible fashion and written in standard English?

Reviewer #4: (No Response)

Reviewer #4: 1) The introduction reiterates “importance of immersion” multiple times. Can be reduced by ~25% without loss of content.

2) Immersion,” “presence,” and “realism” are sometimes used interchangeably. These are not equivalent constructs in VR literature.

3) Fiber optic sensor non-linearity is mentioned but not quantified. Readers may question why closed-loop force control was abandoned.

4) fMRI figures are purely visual. No effect size maps or confidence indicators are shown.

5) Minor grammatical and stylistic issues throughout (e.g., redundant phrases, awkward transitions). Not critical but noticeable.

6) Experiment 1: n = 13 Experiment 2: n = 7,These are very small samples, especially given whole-brain analyses.

Low power inflates effect sizes and false discovery risk. Especially problematic since no correction survives. Re-frame conclusions accordingly. Alternatively, restrict interpretation to within-subject descriptive patterns.

7) Trumpet fingering task uses visually cued button presses while breath input is fixed. This task resembles a button-press paradigm more than natural interaction. The added value of a multi-finger haptic device is not strongly demonstrated over simpler input devices. Need stronger justification for why this task requires multi-finger force feedback or acknowledge task limitations

**Do you want your identity to be public for this peer review?** For information about this choice, including consent withdrawal, please see our For information about this choice, including consent withdrawal, please see our Privacy Policy .

Reviewer #4: **Yes:** Rana Sami Ullah KhanRana Sami Ullah Khan

---

## [Author Response · Author response to Decision Letter 4]

3 Feb 2026

Thank you for your constructive review. Please refer to the 'Response to Reviewers' file for my detailed answers to the comments.

---

## [Editor Report · Decision Letter 4]

4 Feb 2026

Exploring Immersion Through a fMRI-Compatible Multi-Finger Handheld Haptic Display

PONE-D-25-19819R4

Kind regards,

Usman Ghafoor

Academic Editor

PLOS One

Additional Editor Comments (optional):

The paper quality has been sufficiently improved that warrants its publication.
---

## [Editor Report · Acceptance letter]

PONE-D-25-19819R4

PLOS One

Dear Dr. Kim,

I'm pleased to inform you that your manuscript has been deemed suitable for publication in PLOS One. Congratulations! Your manuscript is now being handed over to our production team.

Kind regards,

on behalf of

Dr. Usman Ghafoor

Academic Editor

PLOS One